# Stackelberg Learning with Outcome-based Payment

**Tom Yan**
Carnegie Mellon University
tyyan@cmu.edu

**Chicheng Zhang**
University of Arizona
chichengz@cs.arizona.edu

## Abstract

With businesses starting to deploy agents to act on their behalf, an emerging challenge that businesses have to contend with is how to incentivize other agents with differing interests to work alongside its own agent. In present day commerce, payment is a common way that different parties use to *economically* align their interests. In this paper, we study how one could analogously learn such payment schemes for aligning agents in the decentralized multi-agent setting. We model this problem as a Stackelberg Markov game, in which the leader can commit to a policy and also designate a set of outcome-based payments. We are interested in answering the question: when do efficient learning algorithms exist? To this end, we characterize the computational and statistical complexity of planning and learning in general-sum and cooperative games. In general-sum games, we find that planning is computationally intractable. In cooperative games, we show that learning can be statistically hard without payment and efficient with payment, showing that payment is necessary for learning even with aligned rewards. Altogether, our work aims to consolidate our theoretical understanding of outcome-based payment algorithms that can economically align decentralized agents.

## 1 Introduction

Increasingly, we are seeing businesses deploying agents to carry out tasks on their behalf. In the coming agentic era, we will inevitably have multiple, decentralized agents interacting together. An emerging challenge that businesses may have to face is how to incentivize other agents to work alongside its agent. This challenge requires addressing a central difficulty in decentralized multi-agent systems, which is that of differing interests.

In present day commerce, payment is a standard way that two parties use to resolve this challenge and more closely align their business interests. This inspires us to study the overarching question in this paper: how can we analogously implement such payment schemes in the multi-agent setting and enable *economic alignment*? That is, if I am a business looking to use payment to incentivize another business (and/or its agent) to work with my agent, how can I learn a good policy for my agent along with a payment scheme to go with it?

On a technical level, this setting may be viewed as a Stackelberg Markov game. In this paper, we study the two-player Stackelberg game, where one player (leader) commits to a policy taking into account the best response to the policy by the other player (follower). We focus on Stackelberg Markov games in particular as agents will be interacting over multiple turns and potentially long horizons. Finally, to model the payment aspect, the leader is able to also increase the reward of the follower in the Markov game, which may be viewed as a form of reward shaping in line with the existing formulation in the literature [Ben-Porat et al., 2024, Bollini et al., 2024, Ivanov et al., 2024, Scheid et al., 2024, Wu et al., 2024].

In this work, we aim to consolidate the theoretical foundations of Stackelberg learning with payment, as complexity results have yet to be established for two-player Stackelberg Markov games. We focus

39th Conference on Neural Information Processing Systems (NeurIPS 2025).

| Planning, Learning | Without Payment | With Payment | |
|---|---|---|---|
| | DAG | Tree | DAG |
| Cooperative | ✓, ✗ (Theorem 5.1 5.2) | ✓, ✓ | ✓, ✓ (Theorem 6.4 6.5) |
| General Sum | ✗, ✗ (Proposition 4.1) | ✓, ✓ (Proposition 4.3) | ✗, ✗ (Theorem 4.2) |

Table 1: Planning & learning settings where computationally *and* statistically efficient algorithms exist.

on a fundamental question: is there an efficient algorithm that can provably compute or learn the optimal policy and payment? Indeed, this is an important question to address as businesses in the future would want payment schemes with *provable guarantees*, so as to ensure that their expenditure is optimal.

**Contributions:** We analyze the planning and learning setting through both the computational and statistical lens. Please see Table 1 for an overview of our results.

1. We begin by considering planning in general-sum games. Is there an efficient algorithm that can return the optimal policy and payment? We prove that such a computationally efficient algorithm cannot exist unless NP=P, and identify the structural property of the MDP that results in this hardness. To complement the negative results, we develop an efficient algorithm, applicable when this property is removed.

2. Next, we turn to Cooperative games, which is a broad subclass of Markov games useful for modeling e.g. the interaction between AI service-providers and their users. Moreover, planning is computationally efficient in this setting, making it plausible that efficient learning algorithms may be attainable. As the rewards are already aligned, we begin by considering learning in the Stackelberg game without payment. Surprisingly, however, we find that an efficient algorithm cannot exist, this time in the statistical sense. We identify structural properties of the MDP that result in statistical hardness, and develop an efficient algorithm for when such properties are removed to complement our negative results.

3. Finally, we study learning in Cooperative games with payment. Can payment be used to alleviate the statistical hardness of learning? We answer this in the affirmative by showing that we can adapt existing no-regret RL algorithms to enable sample-efficient learning. In closing, we also use this setting to contrast the two different payment settings we study. We derive matching upper and lower regret bounds for when the leader has to make payments upfront versus on-the-fly, allowing us to quantitatively assess the benefits of being able to make payments on-the-fly.

## 2 Formulation

### 2.1 Stackelberg Markov Game

We consider the standard two-player, episodic finite-horizon Markov game $M$ parameterized by $\langle S, A, B, H, P, \rho, r^L, r^F \rangle$ with state space $S$, initial state distribution $s_0 \sim \rho$, transitions $P$ and episode length $H$. The leader has action set $A$ and reward $r^L \in [-1, 1]$, the follower action set $B$ and reward $r^F \in [-1, 1]$. In the case that the game is cooperative, $r^L = r^F$.

In the problem of online learning for Stackelberg Markov games, the learner plays the role of the leader, where apriori the reward functions $r^L, r^F$ and the transitions are unknown to the leader. At each episode $k \in [T]$, the leader commits first to a policy $\pi_k$. The follower best responds to $\pi_k$ with $\mu(\pi_k) \in \text{argmax}_\mu V^{\pi_k, \mu}(s_0; r^F)$. One may view best response as the equilibrium behavior of the follower to the leader policy.

After the episode, the leader and the follower observe the resultant trajectory $\tau_k = \left\{ (s_i, a_i, b_i, r^L(s_i, a_i, b_i), r^F(s_i, a_i, b_i)) \right\}_{i=1}^{H}$ realized by the chosen policies in $M$: $a_i \sim \pi_k(s_i), b_i \sim \mu(\pi_k)(s_i), s_{i+1} \sim P_i(\cdot|s_i, a_i, b_i)$.

This trajectory is the outcome of the policies' interaction, which in turn determines the *outcome-based* payment the follower receives.

**Leader Payment:** Following existing formulations in prior literature, the leader can increase $r^F$ by creating outcome-based payment $b_i^k(s_i, a_i, b_i)$, if state-actions $s_i, a_i, b_i$ are realized during the episode, $s_i, a_i, b_i \in \tau_k$. This results in a modified Markov game where the leader is able to additionally assign payment, with the payment function having signature $b_i^k : S \times A \times B \to \mathbb{R}^+$.

We note that the outcome-based payment need not correspond to direct monetary transfer. For example, we may be interested in modeling the setting where the leader is an AI-service-provider and the follower is a customer user. The leader spends money to improve its agent, and this improved agent adds additional value (e.g. more saved time) for the user during its use. But during this interaction, there is no direct transfer of money from the company to the user.

Thus, to model indirect payments in addition to direct ones, we introduce a final piece of notation, multiplier $\kappa \in \mathbb{R}^+$. $\kappa \cdot b_i^k(s_i, a_i, b_i)$ corresponds to the proportional cost to the leader in creating payment (reward) $b_i^k(s_i, a_i, b_i)$ for the follower. We believe proportionality is a natural assumption to make, and verily $\kappa = 1$ corresponds to direct payment.

## 2.2 Payment Settings

To complete the formulation, we touch on the two types of payment settings considered in this paper.

**Trajectory Payment:** The first is the existing payment setting commonly studied in prior literature, which we term trajectory payment. Here, a payment is made by the leader for every state-action on the realized trajectory. This form of payment is considered in principal-agent contracting literature, where the trajectory informing how much the leader will be paying ex-post [Dutting et al., 2021].

Moreover, this form of payment corresponds to the trendy outcome-based pricing model, which is experiencing rapid adoption by several notable SaaS companies due to the rising usage of AI agents [Stripe, 2025, Intercom, 2025, Zendesk, 2025]. Indeed, this marks a fundamental paradigm shift in software pricing in industry, moving from seat-based subscriptions (traditional SaaS) and usage-based models (cloud infrastructure) to now outcome-based pricing in the agent era [Boston Consulting Group, 2025, Sequoia Capital, 2025]. This also makes it imperative then to bolster our theoretical understanding of outcome based pricing, which we study in this paper.

**Upfront Payment:** In this paper, we will also consider a setting that we term upfront payment. As the name suggests, the leader pays for every state-action in the MDP, regardless of the realized trajectory. Note that the follower is still paid based on the realized trajectory. This is more realistic in settings where the leader pays indirectly to the benefit of the follower, and is bound by temporal constraints such that the payment cannot be made on-the-fly.

For a motivating example, consider the AI-service provider setting discussed earlier. The company invests before deployment to improve the agent's functionality, which means that the user (follower) gains added value (reward) on the trajectory realized during the agent's use. However, the key temporal constraint is that the company cannot improve its agent on-the-fly, as the users are using it. Thus, this makes upfront payment a more realistic model of the leader's expenditure. The leader had to invest upfront to improve the agent's capabilities in all states, even though this includes off-trajectory states that are not visited during the interaction with the user. For instance, suppose the agent is a computer-using-agent [Anthropic, 2024]. The user may use it to handle emails, and the agent would act in states of the computer corresponding to the inbox. However, even though the company had also invested to improve the agent's capabilities in coding, the user may not invoke the agent to do so (perhaps due to excessive risk). And so, the agent would not have acted in other states of the computer corresponding to the codebase.

More generally, there is sizable body of economics contracting literature studying settings where only ex-ante (upfront) payment is possible. Some reasons for this include non-enforceable contracts, where the principal can renege upon observing the outcome Hart and Moore [1988]. Another cause for this may be non-verifiable outcomes; that is, when outcomes cannot be verified, ex-post contracts become unenforceable as there is no way to condition legally binding payments Aghion and Holden [2011]. Finally, one other reason may simply be that the agent is risk-averse, thus preferring upfront payment in face of stochastic outcomes Laffont and Martimort [2002].

**Leader Optimization:** Putting it all together, we can now write down the resulting Stackelberg game under the two payment settings.

**Definition 2.1.** *In Stackelberg Markov games with trajectory payment, the leader optimizes:*

$$\max_{\pi, b \geq 0} \quad V^{\pi, \mu(\pi)}(s_0; r^L - \kappa \cdot b)$$
$$s.t. \quad \mu(\pi) \in argmax_{\mu'} V^{\pi, \mu'}(s_0; r^F + b) \tag{1}$$

*In Stackelberg Markov games with upfront payment, the leader optimizes:*

$$\max_{\pi, b \geq 0} \quad \left( V^{\pi, \mu(\pi)}(s_0; r^L) - \kappa \cdot \sum_{s, a, b \in S \times A \times B} b(s, a, b) \right)$$
$$s.t. \quad \mu(\pi) \in argmax_{\mu'} V^{\pi, \mu'}(s_0; r^F + b) \tag{2}$$

Before moving on, we highlight the generality of the class of games we are studying. The class of Stackelberg Markov games with payment generalizes Stackelberg Markov games. Indeed, constraining the leader to zero payment (i.e. $(\pi, b) = (\pi, 0)$) corresponds to the leader's policy space in Stackelberg Markov Games. Analogously, for Cooperative Stackelberg Markov games with payment studied in the later sections, this class of games generalizes Cooperative Stackelberg Markov games.

## 3  Related Works

As we focus on Stackelberg Markov games with payment, our paper is most related to two lines of work. The first is the line of work studying the complexity of Stackelberg policy computation in Markov games. And the second is algorithms for computing optimal payment schemes in MDPs. We cover both lines of work below, and include a discussion on additional related works in Appendix E.

**Stackelberg Optimal Policies in Markov Games without Payment:** Due to the wide applicability of the Stackelberg Markov games, there has been a long line of work seeking to understand how to compute optimal leader policies with provable guarantees.

For planning, Conitzer and Sandholm [2006], Letchford et al. [2012], Letchford and Conitzer [2010] study the computational tractability of optimal Stackelberg policy computation in Markov games and subclasses thereof. For stochastic MDPs, they establish that computing the optimal Stackelberg policy is NP-Hard.

For learning, Zhao et al. [2023] studies the statistical complexity in cooperative bandit games. Bai et al. [2021] studies the statistical complexity in bandit-RL games, a particular subclass of Markov games. Our work differs from this line of work in focusing on Markov games, which are more general than bandit-RL games and have longer horizon than bandit settings. Moreover, the leader is allowed to use payments to shape the follower's rewards. As we will see, this turns out to be crucial for improved exploration during learning in certain settings.

**Learning the Optimal Payment Scheme in MDPs:** Recently, there has been burgeoning interest in computing optimal payment schemes for contracting agents to act in MDP environments, wherein the leader may increase the follower's rewards as a form of reward shaping to incentivize the follower to play policies desirable to the leader.

The single-agent MDP setting, where only the follower acts in the MDP and the leader incentivizes, is formulated by Ben-Porat et al. [2024], Chen et al. [2022]. This is followed by a series of interesting work by Bollini et al. [2024], Ivanov et al. [2024], Wu et al. [2024], studying learning under a variety of different payment functions taking as input the state, the state-action or the state-next-state. Our work adds to this line of work by focusing on two-player Markov games, which generalize the single-player setting. Furthermore, while previous works mostly focus on trajectory payment, we also consider upfront payment, applicable in settings where the leader cannot pay on the fly due to temporal constraints. We derive tight regret guarantees to contrast the two differing payment settings.

The paper closest in formulation to that of ours is that by Scheid et al. [2024], who considers the same state-action based payment function in the bandit setting. Our work differs in focusing on Markov games, with a longer horizon than that in bandit settings. This in turn introduces difficulty in terms of exploration, and requires a more nuanced optimal payment computation beyond the binary search approach used in [Scheid et al., 2024].

Finally, as payment may be viewed as strategic reward shaping, our analysis is also related to existing RL literature that seeks to theoretically quantify the benefits of reward shaping [Ng et al., 1999]. Gupta et al. [2022] quantifies how statistical sample complexity is improved by reward shaping in the single-agent setting. By contrast, in our work, we study improved sample complexity in two-player cooperative Stackelberg Markov games.

# 4 Planning in General-sum Games

In this section, we ask: is there an efficient algorithm that can compute the optimal policy and payment in general-sum games? We investigate the computational complexity of such an algorithm, starting with the planning setting, where the Markov game dynamics and reward functions are known.

Our main finding is that there is no such computationally efficient algorithm unless NP=P. Outcome-based payment does not alleviate the computational intractability of computing the optimal Stackelberg policy, even in planning [Conitzer and Sandholm, 2006]. We identify that when the MDP has DAG structure, this leads to computational intractability. Later in the section, we complement this negative result with a positive result for when the MDP has tree structure. All proofs in this section may be found in Appendix A.

## 4.1 Hardness Results

We first derive a result showing that it is NP-Hard to compute the optimal leader policy even in deterministic MDPs, without payment. Note that in [Conitzer and Sandholm, 2006], computational intractability is demonstrated in stochastic MDPs.

**Proposition 4.1.** *Under Markov games that are deterministic DAGs, it is NP-Hard to compute the optimal policy:*

$$\max_{\pi} \quad V^{\pi,\mu(\pi)}(s_0; r^L)$$
$$s.t. \quad \mu(\pi) \in argmax_{\mu'} V^{\pi,\mu'}(s_0; r^F)$$

Helpfully, deterministic MDPs allow us to provide guarantees for both two payment settings. As we show in the proof, the optimal payment scheme pays zero in off-policy states, which can be readily characterized in deterministic MDPs. This result is intuitive as paying in off-policy states only incentivizes the follower to deviate off-policy, which is undesirable and increases leader total payment. With this result, we can derive that the optimal payment scheme is the same under trajectory and upfront payment. Thus, we use same construction, which provides a reduction to the PARTITION problem, to prove computational intractability under both payment settings.

**Theorem 4.2.** *Under Markov games that are deterministic DAGs, it is NP-Hard to compute the optimal policy and optimal trajectory payment:*

$$\max_{\pi,b \geq 0} \quad V^{\pi,\mu(\pi)}(s_0; r^L - \kappa \cdot b)$$
$$s.t. \quad \mu(\pi) \in argmax_{\mu'} V^{\pi,\mu'}(s_0; r^F + b)$$

*and it is also NP-Hard to compute the optimal policy and optimal upfront payment:*

$$\max_{\pi,b \geq 0} \quad \left( V^{\pi,\mu(\pi)}(s_0; r^L) - \kappa \cdot \sum_{s,a,b \in S \times A \times B} b(s,a,b) \right)$$
$$s.t. \quad \mu(\pi) \in argmax_{\mu'} V^{\pi,\mu'}(s_0; r^F + b)$$

In closing, we note that the optimal objective value of the subset of Markov games used to reduce to the PARTITION problem is an integral multiple of $1/2$. Due to this, we have that computational intractability in planning implies computational intractability in learning. In more detail, let $M^*$ be the optimal objective value, which is an integer multiple of $1/2$. Suppose by contradiction that we had an algorithm with sublinear regret $T^\alpha$ ($\alpha < 1$). We can then set $T$ large enough such that $T^\alpha/T < 1/2$. This allows us to infer $M^*$ exactly by rounding to the nearest $1/2$, giving us a computationally efficient algorithm for answering the decision version of the PARTITION problem, which is a contradiction.

---

**Algorithm 1** Planning Algorithm for MDP with Deterministic Tree Structure

---

**Require:** Pre-computed policy $\pi^{-} \in \text{argmin}_{\pi} V^{\pi, \mu(\pi)}(s_0; r^F)$ (efficiently computed via Nash-VI)
    **for** all root to leaf paths $\tau = s_1, a_1, b_1, s_2, a_2, b_2, ..., s_H, a_H, b_H$ **do**
        Define $\pi(s_i) = a_i$ for $s_i, a_i \in \tau$. In every other state $s_i' \notin \tau$, let $\pi(s_i') = \pi^{-}(s_i')$.
        Compute $\mu(\pi)$ and compute follower Q-values, $Q^{\pi, \mu(\pi)}(\cdot, \cdot, \cdot)$.
        Solve for the minimal payment scheme using LP:

$$b^{\tau}(\pi) = \text{argmin}_b \sum_{s_i, a_i, b_i \in \tau} b(s_i, a_i, b_i)$$

$$\text{s.t.} \sum_{i \geq h, s_i, a_i, b_i \in \tau} r^F(s_i, a_i, b_i) + b(s_i, a_i, b_i) \geq \max_{b_h' \neq b_h} Q^{\pi, \mu(\pi)}(s_h, a_h, b_h'; r^F)$$

$$(3)$$

    **end for**
    Output the leader policy $\pi$ and payment scheme of the path $\tau$ with maximal return $\sum_{s_i, a_i, b_i \in \tau} r^L(s_i, a_i, b_i) - \kappa \cdot b^{\tau}(\pi)$.

---

## 4.2 Positive Results

To complement our negative results, we show that positive results are attainable in MDPs without DAG structure. That is, in general-sum games where the MDP has tree structure, there is a polynomial-time algorithm for learning the optimal leader policy and payment. We describe our planning algorithm, Algorithm 1 that forms the crux of our approach to learning in this setting, and is applicable under both trajectory and upfront payment.

**Proposition 4.3.** *Under Markov games that are deterministic trees, there exists a polynomial-time planning algorithm that computes the optimal policy and payment.*

**Remark 4.4.** *To complete the result, we note in Appendix A that there is a simple exploration strategy using payment for general-sum, deterministic trees, as exploration needs to only recover rewards. This strategy allows us to reduce learning to planning, and then apply Algorithm 1.*

Before moving on, we note that in this general-sum game, the leader behaves in a zero-sum like manner in off-policy states in Algorithm 1. This incentivizes the follower to take the desired policy and allows the leader to minimize the total payment needed to incentivize such policy.

Finally, due to the intractability of computing a global Stackelberg optimum, it is natural to consider computing a local Stackelberg optimum instead, so that the policy and payment scheme does attain some guarantees. Building on existing results on first order methods in Stackelberg games [Shen et al., 2024], we derive a first order approach to this end. Note that while our paper is concerned with global Stackelberg optimality guarantees, we use this to illustrate that a more relaxed solution concept can be computed, if desired.

## 5 Learning in Cooperative Games without Payment

The computational intractability in the general-sum case prompts us to investigate whether efficient algorithms are attainable in significant subclasses of Markov games. Cooperative games are a broad subclass of Markov games useful for modeling e.g. the aforementioned AI-service based setting. Indeed, since the goal of the assistant agent is to aid the user, their rewards are aligned. And so, such settings correspond to a two-player cooperative game, making it an important subclass of Markov games to understand.

Moreover, on a technical level, it seems that there is hope for efficient algorithms as planning is efficient in cooperative games (e.g. via Nash-VI as in Bai and Jin [2020]). And so, in this section, we study the question: is there an efficient learning algorithm in cooperative games? We delve into this by first considering cooperative games without payment, which has yet to be addressed in the prior literature. Since the rewards are already aligned, we might expect that there are efficient learning algorithms. To our surprise, however, we find that learning in Cooperative Markov games

can be prohibitively hard, this time in the statistical sense. All proofs in this section may be found in Appendix B.

**Structural properties of MDP:** We identify the specific MDP properties under which exploration can be statistically intractable, along with complementary positive results. In a nutshell, we find that if the MDP has deterministic tree structure, then efficient algorithms are possible. However, allowing for stochastic or DAG transitions leads to statistical hardness.

**Theorem 5.1.** *There exists a turn-based Stochastic Tree Markov game such that: any (possibly randomized) algorithm that returns the optimal leader policy with probability at least $1/2$ requires at least $\Omega(2^{|S|})$ number of episodes.*

**Theorem 5.2.** *There exists a turn-based Deterministic DAG Markov game such that: any (possibly randomized) algorithm that returns the optimal leader policy with probability at least $1/2$ requires at least $\Omega(2^{|H|})$ number of episodes.*

**Proposition 5.3.** *Under Markov games that are deterministic trees, then there exists a polynomial-time algorithm that can learn a near-optimal leader policy.*

We remark that the statistical intractability results are based on a "needle-in-the-haystack" construction, where only a specific combination of leader actions is optimal. Structural properties of the MDP like stochastic or DAG transitions allow us to embed this construction in the MDP. Combined with the follower best responding instead of coordinating exploration with the leader, we can show that an exponential number of samples is needed by the leader to find the right combination, even if the rewards are already aligned.

**Relaxing Follower Best Response behavior:** As the statistical hardness is due to both the structural property of the MDP and the best response nature of the follower, a natural question one may ask is: can relaxing the latter alleviate statistical hardness and allow for efficient learning across all MDPs?

The natural way to relax best response is to consider best response under $\lambda$-entropy-regularization, which generalizes follower best response (corresponding to when $\lambda = \infty$). This behavior model is often used to model human behavior in human-AI interaction and behavioral economics literature [Ziebart et al., 2010, Reddy et al., 2018, McKelvey and Palfrey, 1995]. However, we again find that learning with this follower behavior does not allow for more sample efficient exploration:

**Theorem 5.4.** *There exists a turn-based Deterministic DAG Markov game such that: any (possibly randomized) algorithm that outputs the optimal policy given $\lambda$-Entropy-regularized best response with probability at least $1/2$ requires at least $\Omega(\exp(\lambda^2 H/8))$ episodes if $\lambda \le 1$ and $\Omega(\exp(H/8))$ episodes if $\lambda > 1$.*

In closing, we offer a conceptual interpretation of the technical results in this section, using the example of the assistant agent and the user. Our results suggest that the service provider company can have difficulty exploring, due to the user's best response. Indeed, users are simply looking to use the agent wherever it is at its best, and will not use the agent for the sake of its improvement. In particular, this means that users are not willing to use the agent in states that it currently does not currently excel in. Even though, these are precisely the states that the agent needs to obtain more training samples in. And so, this suggests that if the company wants to efficiently explore to learn an even better agent, incentivized exploration is needed.

# 6 Learning in Cooperative Games with Payment

In sum, we know from the previous section that in Stackelberg games, coordinated exploration is necessary for efficient learning. And so, in this section, we study how payment can be used to align the follower and enable efficient leader exploration. Our overall finding is that payment can lead to efficient exploration, and alleviate the statistical hardness in cooperative games without payment. All proofs in this section may be found in Appendix C.

## 6.1 Regret Guarantees in Cooperative Games

We study regret guarantees under the standard reinforcement learning setup with unknown transitions and unknown rewards, which can be stochastic.

**Learning protocol:** At each episode $k \in [T]$, the leader commits first to a policy $\pi^k$ and a payment function $b^k$. The follower best responds to $\pi^k$ with $\mu(\pi^k) \in \text{argmax}_\mu V^{\pi_k, \mu}(s_0; r^F + b^k)$. After the episode, the leader and the follower observe the resultant trajectory $\tau_k = \left\{(s_i, a_i, b_i, r^L(s_i, a_i, b_i))\right\}_{i=1}^H$ realized by the chosen policies in $M$ (recall that $r^L = r^F$). The goal of the learner is to minimize its Stackelberg regret, defined as follows:

**Definition 6.1.** *In Stackelberg games with trajectory payment, the Stackelberg regret is defined as:*

$$\mathcal{R}(T) = \sum_{k=1}^T V^{\pi^*, \mu(\pi^*; r^F + b^*)}(s_0; r^L - \kappa \cdot b^*) - V^{\pi^k, \mu(\pi^k; r^F + b^k)}(s_0; r^L - \kappa \cdot b^k)$$

*The regret under upfront payment regret may be defined analogously.*

Towards analyzing Stackelberg regret, we characterize the optimal policy and trajectory payment when $r^L = r^F$; we can analogously show the same result under upfront payment.

**Lemma 6.2.** *For any $\pi^*, b^*$ such that:*

$$
\begin{aligned}
\pi^*, b^* = \text{argmax}_{\pi, b} \quad & V^{\pi, \mu(\pi; r^F + b)}(s_0; r^L - \kappa \cdot b) \\
\text{s.t.} \quad & \mu(\pi; r^F + b) \in \text{argmax}_{\mu'} V^{\pi, \mu'}(s_0; r^F + b)
\end{aligned}
\tag{4}
$$

*If $r^L = r^F$, then we must have $\pi^*, \cdot = \text{argmax}_{\pi, \mu} V^{\pi, \mu}(s_0; r^L)$ and $b^* = 0$.*

With this, we have that the optimal payment scheme in any cooperative game must be zero, as one would intuitively expect with already aligned rewards. This allows us to decompose Stackelberg regret into regret due to sub-optimality in policy and regret due to payment used during exploration, which will be responsible for the differing rates between trajectory and upfront payment.

Moreover, we note an interesting contrast due to this result. As we just saw, learning can be prohibitively hard in the absence of payment. Hence, we have that payment is not necessary in planning, but is crucial for learning (efficiently).

The crux of our positive results is that we can apply the canonical optimism under uncertainty principle to achieving sublinear Stackelberg regret. This follows from the observation that payment enables optimism in learning, which the leader can operationalize by setting payments according to its bonuses. This incentivizes the follower to also explore optimistically. A key lemma for bounding the policy regret portion of Stackelberg regret goes as follows.

**Lemma 6.3.** *Suppose we can construct an optimistic MDP $M_k$ of the true MDP $M$. Let the optimal leader policy under $M_k$ be $\pi_k$, then:*

$$\sum_{k=1}^T V_M^{\pi^*, \mu_M(\pi^*)}(s_0; r^L) - V_M^{\pi^k, \mu_M(\pi^k)}(s_0; r^L) \leq \sum_{k=1}^T V_{M_k}^{\pi^k, \mu_{M_k}(\pi^k)}(s_0; r^L) - V_M^{\pi^k, \mu_{M_k}(\pi^k)}(s_0; r^L)$$

Note that because the leader knows $M_k$, they know the policy $\mu_{M_k}(\pi^k)$ that they would like to incentivize the follower to play. Using this, we show that one can also bound the regret due to the cumulative payment, to obtain the following regret guarantees.

**Theorem 6.4.** *UCB-VI-FP (Algorithm 2) incurs $O(T^{1/2})$ regret under trajectory payment. This is tight as there exists a subset of Markov games, where any learning algorithm must incur $\Omega(T^{1/2})$ regret.*

**Theorem 6.5.** *There exists an algorithm, leveraging UCB-VI-FP as subroutine, that incurs $O(T^{2/3})$ regret under upfront payment.*

## 6.2 Contrasting Trajectory Payment with Upfront Payment

Finally, as positive results are attainable in Cooperative Markov games, we can analyze the difference in regret rates under the two different payment settings. What is the benefit afforded by settings where the leader can pay on-the-fly? Towards answering this question, we analyze the simple setup of unknown, deterministic rewards. Helpfully, this learning task already a sizable contrast in terms of regret between the two settings. We provide tight bounds on regret guarantees under both payment settings to contrast the two payment settings.

---

**Algorithm 2** UCB-VI with Follower Payment (UCB-VI-FP)

---

Initialize $Q_h(s,a,b) = H$ for all $h \in [H], s,a,b \in S_h \times A \times B$.
**for** $k = 1, ..., T$ **do**
    **for** $h = H, ..., 1; s,a,b \in S_h \times A \times B$ **do**             ▷*construct* $M_k$
        Compute estimated transitions from data in buffer: $\hat{P}_h(s'|s,a,b) = \frac{N_h^k(s,a,b,s')}{N_h^k(s,a,b)}$
        Compute optimistic rewards of $M_k$ from reward samples in buffer: $\hat{r}_h^k(s,a,b) = \bar{r}_h^k(s,a,b) + c\sqrt{\frac{H^2}{N_h^k(s,a,b)}}$           ▷*standard bonus for stochastic rewards*
        $Q_h(s,a,b) = \min(H, \hat{r}_h^k(s,a,b) + \hat{P}_h^k V_{h+1}(s,a,b))$
        $V_h(s) = \max_{a,b} Q_h(s,a,b)$
    **end for**
    Leader commits to Stackelberg policy $\pi^k$: $\pi^k(s_h) = \text{argmax}_a \max_b Q_h(s_h,a,b)$.
    Set outcome-based payment scheme: $\beta_h^k(s_h,a_h,b_h) = 2 \cdot c\sqrt{\frac{H^2|S|}{N_h^k(s,a,b)}}$.
    **for** $h = 1, ..., H$ **do**
        Leader plays $a_h^k \sim \pi^k(s_h^k)$, follower plays $b_h^k$ via $\mu(\pi^k)$
        Transition to $s_{h+1}^k \sim P(\cdot|s_h^k,a_h^k,b_h^k)$ and save data $(s_h^k,a_h^k,b_h^k,s_{h+1}^k)$ in buffer
    **end for**
**end for**

---

**Proposition 6.6.** *UCB-VI-FP with indicator bonus incurs constant $O(|S||A||B|)$ regret under trajectory payment, where we designate reward under indicator bonus to be $\hat{r}(s,a,b) = 1\{$if $(s,a,b)$ is unvisited$\}$ and $r(s,a,b)$ o.w.*

As the regret bound is constant in $T$, we have that the bound must be tight. Next, we derive regret rates under upfront payment, whose regret lower bound requires a significantly nuanced probabilistic argument using Yao's lemma.

**Proposition 6.7.** *There exists an algorithm, leveraging UCB-VI-FP with indicator bonus as subroutine, that incurs $O(T^{1/2})$ regret under upfront payment.*

**Proposition 6.8.** *There exists a subset of Markov Game instances such that any learning algorithm has to incur $\Omega(T^{1/2})$ regret under upfront payment.*

The construction of the negative result reveals the key difference in two payment schemes. In a nutshell, upfront payment is affected by difficult-to-reach states ($\epsilon$-significant states [Jin et al., 2020]). On the other hand, trajectory payment is unaffected as the payment is made only if the follower does reach such a state. That is, the leader's payment for actions in that statement is weighted by the visitation probability.

And so, the key difficulty in exploration under upfront payment is that when payment is needed to incentivize the follower to reach insignificant states, a lot of the payment can be wasted even if the follower is aligned, due to the low visitation probability. This is directly responsible for the sizable change in the regret guarantee, going from $O(1)$ to $\Omega(T^{1/2})$. Overall, this suggests that if the leader cannot pay on-the-fly, the payment scheme should factor in the reachability of states.

## 7 Discussion

In this work, we study learning in Stackelberg Markov games with payment. To consolidate the theoretical foundations of this setting, we chart the computational and statistical complexity of both planning and learning.

**Future Work:** Due to the intractability of general-sum settings, we believe that there is much more work to be done in analyzing more specific subclasses of Markov games. Which other subclasses of Markov games are such that efficient algorithms are attainable?

**Limitations:** One key assumption underlying our paper is that the follower's action can be observed by the leader. We believe that this can be realistic for modeling certain digital settings (such as computers), wherein the agent's actions can be readily tracked (computer-using-agent's actions can be logged and monitored) [Anthropic, 2024, Sumers et al., 2025]. With that said, handling the case

for when the follower's action is not observable is very important, especially in physical environment where monitoring is not possible. And we believe that results from the setting we study can serve as a stepping stone towards results in partial information settings with unobserved actions.

Another key underlying assumption is that the leader can readily observe the follower's reward, either directly or through the follower's report. It is conceivable that in cases the leader cannot observe the reward directly, the follower may not report their reward truthfully. In such settings, we note two observations. Let $(\pi^*(r), b^*(r))$ denote an optimal policy under reported follower reward $r$. Let $r^F$ denote the true reward and $r'^F$ the reported reward.

First, if we are in the cooperative setting, we observe that there is no incentive for the follower to misreport. Because the leader payment is zero, truthful reporting yields the highest return: $V^{\pi^*(r^F),\mu(\pi^*(r^F))}(s_0; r^F) \geq V^{\pi^*(r'^F),\mu(\pi^*(r'^F))}(s_0; r^F)$.

Second, in the general-sum bandit setting with direct payment considered by Scheid et al. [2024], the payment can now be nonzero but the follower's gain from misreporting is bounded.

**Proposition 7.1.** *Suppose the follower can misreport $r^F$ up to $\Delta$, $\|r'^F - r^F\|_1 \leq \Delta$. In the bandit setting, the follower's return can change by at most:*

$$|V^{\pi^*(r^F),\mu(\pi^*(r^F))}(s_0; r^F + b^*(r^F)) - V^{\pi^*(r'^F),\mu(\pi^*(r'^F))}(s_0; r^F + b^*(r'^F))| \leq 2\Delta$$

*and the leader's return can change by at most:*

$$|V^{\pi^*(r^F),\mu(\pi^*(r^F))}(s_0; r^L - b^*(r^F)) - V^{\pi^*(r'^F),\mu(\pi^*(r'^F))}(s_0; r^L - b^*(r'^F))| \leq 2\Delta$$

However, an open question is whether such a bound carries over to the Markov game case. How much could the follower gain from misreporting $r^F$ up to $\Delta$? Are there algorithms that can induce truthfulness, while still attaining some optimality guarantees? We believe there is a fruitful line of work to be done to handle cases where the leader cannot directly observe and/or verify the follower rewards.

# 8  Acknowledgement

TY is grateful for the support of the NSF GRFP. CZ is supported by NSF Award IIS-2440266.

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
