# OpenReview forum: "Stackelberg Learning with Outcome-based Payment"
_NeurIPS.cc/2025/Conference — NeurIPS 2025 poster_

### Official Review · Reviewer_jtyD · 2025-07-01

**Clarity:** 1
**Significance:** 3
**Originality:** 2
**Rating:** 3
**Confidence:** 3

**Summary:**

This paper provides an algorithm for two-stage resource allocation problem modelling two-player leader-follower task assignment within the context of a Stacklberg game. The authors provide an algorithm to solve this game. The authors show that efficient optimal planning in general-sum games is impossible unless NP=P, while identifying structural conditions that allow tractable solutions. Furthermore, in cooperative games without payments, learning is statistically hard despite aligned rewards, but tractable under certain conditions. With payments, sample-efficient learning becomes possible, and comparing upfront versus on-the-fly payments provide advantages.

**Questions:**

- How did the structure of the payment(s) and reward(s) change in the Stackelberg game to turn this from a cooperative game to general sum game?

**Ethical Concerns:**

["NO or VERY MINOR ethics concerns only"]

**Final Justification:**

While the empirical results strengthen the paper, the clarity of formulations and proofs needs improvement - its a bit hard to verity. Furthermore, I feel that a lot of literature that should have been covered in the literature review were not. Leads me to think that some similar concepts may already exist in the field - and could be leveraged, compared against, or benchmarked against.

I see that the other reviewers do appreciate the work more than myself. I'm happy to not be a blocker on this front, perhaps there's something I've missed. I've raised my score, but the recommendation the same - perhaps under revisions it could become a better work.

**Limitations:**

- The construct of the Stackelberg game the authors propose - task for payment, refers really to only one type of game, and not a general class of Stackelberg games. This is fair, but limited to this setting.

- Overall, with the lack of empirical results, and lack of baseline comparison with many other approaches to solving Stackelberg games, I do not feel this paper contributes significantly to advancement in Stackelberg game learning.

**Paper Formatting Concerns:**

- Proof sketches should be included in the main paper for the reader to get a general idea how the proof for each theorem was arrived at.
- Open question (Prop. 7.1) in the final section is generally not appropriate for a full paper.

**Quality:**

2

**Strengths And Weaknesses:**

Strengths:

- The study of Stackelberg-Markov games is in general highly relevant to RLHF and robust human-AI interaction environments. This research has potential to shed insight in the area.

Weaknesses:

- No empirical results to support the author's theoretical claims.
- The complexity theory is somewhat trvial and not meaningful. Stackelberg games are notoriously inefficient to solve for equilibrium. Established bilevel optimization algorithms such as KKT reformulations, which can compute Stackelberg solutions under polynomial time, are not covered in the literature review. The authors need to do more review on SoTA and established Stackelberg solvers.

---

> ### Author Rebuttal · Authors · 2025-07-31
>
> Thanks for your review, reviewer jtyD! We will be sure to incorporate your feedback in our next revision.
>
> ```
> This paper provides an algorithm for two-stage resource allocation problem modelling two-player leader-follower task assignment within the context of a Stacklberg game.
> ```
>
> Just as a point of clarification, our paper is not about “resource allocation” nor “task assignment”. It considers the general Stackelberg Markov Game setting, wherein the leader is able to also set outcome-based payment.
>
> ```
> No empirical results to support the author's theoretical claims.
> ```
>
> Thanks for this note! Indeed, our contributions in this paper are mainly theoretical. Theory is important in this setting as algorithms are used to compute payment. Thus, provable guarantees are important to ensure that the payment computed are sensible/optimal (L39-40).
>
> With this said, we have performed experiments, following your feedback that experiments would further improve the paper. Please find below a description of the experiment we ran.
>
>
> **Setup:** We consider a turn-based Markov Game, where the leader is solving a RL problem and the follower solves a bandit problem in its BR (Bai et al). This class of Markov Games includes the hard instance construction in Section 5. Our goal is to examine whether learning without payment can get stuck in more "average (and not worst) case" Markov Games.
>
>
> The leader is doing RL in a toy MDP with $H = 5$. For the follower, arm $a_1$ leads to a MDP, whose optimal return is the optimal return in the (turn-based) Markov game. Arm $a_2$ has a deterministic high reward that is $\alpha$ that of the optimal return.
>
>
> By varying $\alpha$, we can make the follower get "stuck" in myopically choosing $a_2$, thus preventing the leader from exploring and learning the actual optimal policy. This is the intuition behind the negative results, Theorem 5.1 and 5.2, where we set $\alpha$ very high.
>
>
> For the baseline, we use the single-agent learning algorithm UCB-VI in the without payment case, and compare it against UCB-VI-FP in the with payment case. We track the cumulative regret of the two learning algorithms over $40000$ episodes and across $20$ runs.
>
>
> **Finding:** We experiment with different $\alpha$’s, finding that learning without payment can get stuck in the myopic optimum even when $\alpha$ is as low as $0.5$. Interestingly, this suggests that there are “non-worst-case” Markov games, where exploration can be difficult without payment.
>
>
> Under $\alpha = 0.5$, our plots show that:
>
> 1. In absence of payment, even if the leader is using a (one-sided) no-regret algorithm (UCB-VI), the leader may not be able to explore adequately and incur linear regret.
> 2. UCB-VI-FP attains sublinear regret, showing the importance of payment in incentivizing  exploration needed to learn the optimal policy.
> 3. UCB-VI-FP initially incurs a higher regret, which we expect due to the additional payment used by the algorithm to incentivize exploration. Over time, its regret improves due to exploration shrinking the policy regret, and reduced incentivization and thus payment regret. UCB-VI-FP’s regret eventually dips below that of UCB-VI, with a crossover point at around episode 13500.
> 4. UCB-VI-FP’s regret converges like $O(\sqrt{K})$, which matches our upper bound. We verify this upper bound for every episode $i$ for $i \in [40000]$.
>
> Here is a table summarizing average regret (cumulative regret / $T$) across episodes, averaged over the runs.
>
> | Episode $T$  | $5000$ | $10000$ | $15000$  | $20000$ | $25000$ | $30000$ | $35000$ | $40000$ |
> |--------------|--------|---------|----------|---------|---------|---------|---------|---------|
> | UCB-VI Average Regret  | 0.42   | 0.42    | 0.42     | 0.42    | 0.42    | 0.42    | 0.42    | 0.42    |
> | UCB-VI-FP Average Regret  | 0.70 | 0.49    | 0.40     | 0.35    | 0.31    | 0.28    | 0.26    | 0.24    |
>
>
>
>
> ```
> The complexity theory is somewhat trvial and not meaningful. Stackelberg games are notoriously inefficient to solve for equilibrium. Established bilevel optimization algorithms such as KKT reformulations, which can compute Stackelberg solutions under polynomial time, are not covered in the literature review. The authors need to do more review on SoTA and established Stackelberg solvers.
> ```
>
> Thanks for this comment! We have a few points of disagreement with this characterization of our work:
>
> Firstly, Theorem 4.2 of our results actually show that general-sum Stackelberg Markov games with payment cannot be solved in polynomial time, contrary to your claim that bi-level optimization algorithms “can compute Stackelberg solutions under polynomial time”.
>
> Secondly, bi-level optimizers can compute a local but not global optimum, which is the main goal of our paper. On the topic of local optimum, we did consider adopting an existing optimization approach (Shen et al) to compute this, should it be of interest. Although, as we write in L205-207, solving for the local optimum is not the main focus of our paper.
>
> Thirdly, we believe bilevel optimization algorithms are mainly applicable in planning, where one wishes to compute the optimal policy offline under known rewards and dynamics. On the other hand, we study *learning* in this work as well, where online adaptive exploration is needed to minimize regret under unknown rewards and dynamics. To this end, we have shown several results in the learning setting, where we do not believe bi-level optimization solvers are applicable.
>
> Finally, we will be sure to include this discussion on bi-level optimization solvers and global vs local optimum in our next revision. Thank you for drawing attention to this with your comment! And please let us know if there are any other relevant works on “SoTA and established Stackelberg solvers”, as we believe we have covered all the pertinent works on computing and/or learning *global optimum* in Stackelberg Markov Games in Section 3.
>
>
> ```
> How did the structure of the payment(s) and reward(s) change in the Stackelberg game to turn this from a cooperative game to general sum game?
> ```
>
> In terms of rewards, by definition, the only difference between the general-sum and the cooperative setting is that in the latter case, $r^L = r^F$ (L67). In terms of payments, we show in Lemma 6.2 that the optimal payment in cooperative games is zero.
>
> ```
> The construct of the Stackelberg game the authors propose - task for payment, refers really to only one type of game, and not a general class of Stackelberg games. This is fair, but limited to this setting.
> ```
>
> To clarify, we allow for any Stackelberg Markov game in our setup. Please see the first sentence in our formulation (L64-66), where we make no assumptions on the Stackelberg Markov game we consider besides bounded rewards (a standard assumption in RL theory).
>
> In our setup, the leader is able to also set payment. Thus, Stackelberg Markov games with payment generalizes Stackelberg Markov games. Indeed, constraining the leader to zero payment (i.e. $(\pi, b) = (\pi, 0)$) corresponds to the leader’s policy space in Stackelberg Markov Games without payment.
>
> Altogether, we believe we study a fairly general setup, which is more general than the class of Stackelberg Markov games. We will be sure to make this clear in the revision, and please let us know if there are more concerns we could clarify, thanks!
>
> ```
> Proof sketches should be included in the main paper for the reader to get a general idea how the proof for each theorem was arrived at.
> ```
>
> Thanks for this! We will be sure to include proof sketches with the additional space provided in the revision. Please understand our paper was fairly constrained in terms of space in order to fit all the results in.

---

> > ### Author Response · Authors · 2025-08-06
> > **Reply to Reviewer jtyD**
> >
> > Just to follow up on our rebuttal from a week ago, we wanted to reach out and ask if we have addressed your questions, Reviewer jtyD. Could you please consider letting us know when you get a chance? Thank you for taking the time!

---

> > ### Comment · Reviewer_jtyD · 2025-08-06
> >
> > I thank the authors for diligently responding to my concerns.
> >
> > ---
> >
> > If I'm not mistaken, the payment mechanism appears to be a modification of the original reward scheme (e.g., [1]), which is not explicitly discussed in the literature review of this paper. Furthermore, the payment system (lines 90–93) resembles a principal-agent task-assignment problem, where a principal contracts an agent to act on their behalf to achieve optimal outcomes. This game has been extensively studied in economics and game theory literature [2], yet it is not acknowledged in this paper.
> >
> > The authors' perspective is interesting, though some similar concepts may already exist in the field.
> >
> > ---
> >
> > To clarify further, regarding the KKT reformulation of bilevel optimization problems [3]—a well-known and elegant methodology in bilevel and multi-level optimization—there are known limitations. However, some bilevel optimization problems can be formulated as linear programs (LPs) or convex quadratic programs (QPs), leading to a poly(T) solution. Thus, I am uncertain about the author's claims of NP-hardness (though I could be mistaken). Contrary to the authors' rebuttal, this is a global solution (not local). Additionally, gradient ascent algorithms for bilevel optimization can lead to sufficient local solutions, but I am unsure if this was covered in the literature review [4,5]—it seems relevant here.
> >
> > ---
> > I appreciate the authors' inclusion of additional empirical results. Given the paper's claims, empirical validation is essential to strengthen its credibility and applicability. Overall, I believe the clarity of the work needs improvement. I found the formulations and proofs difficult to follow—either due to my own failure to appreciate the work or because the paper requires further refinement.
> >
> > I'll updated my score in light of the rebuttal and have no further questions.
> >
> > ---
> > [1] Skalse, Joar, et al. "Defining and characterizing reward gaming." Advances in Neural Information Processing Systems 35 (2022): 9460-9471.
> >
> > [2] Dütting, Paul, et al. "The query complexity of contracts." arXiv preprint arXiv:2403.09794 (2024).
> >
> > [3] Dempe, Stephan, and Alain B. Zemkoho. "On the Karush–Kuhn–Tucker reformulation of the bilevel optimization problem." Nonlinear Analysis: Theory, Methods & Applications 75.3 (2012): 1202-1218.
> >
> > [4] Thoma, Vinzenz, et al. "Contextual bilevel reinforcement learning for incentive alignment." Advances in Neural Information Processing Systems 37 (2024): 127369-127435.
> >
> > [5] Naveiro, Roi, and David Ríos Insua. "Gradient methods for solving stackelberg games." International conference on algorithmic decision theory. Cham: Springer International Publishing, 2019.

---

> ### Author Response · Authors · 2025-08-07
> **Reply to Reviewer jtyD**
>
> Thank you for getting back to us, Reviewer jtyD! We appreciate it, and for completeness, we would like to follow up on some of the points raised.
>
> ```
> If I'm not mistaken, the payment mechanism appears to be a modification of the original reward scheme (e.g., [1]), which is not explicitly discussed in the literature review of this paper.
> ```
>
> To clarify, payment in our setting corresponds to reward shaping, which dates back to [Ng et al 1999]. We explicitly discuss the literature review of this first in L33-35 and then in L150-154.
>
> [1] looks to be about reward gaming, and characterizes what happens when a single (fixed) imperfect reward is optimized instead of the true reward. In our setting, the leader gets to adaptively modify the true reward over episodes to incentivize exploration. And so, the two formulations seem to be somewhat different on our first reading. We plan to take another closer look, and please let us know if there is a connection we missed here, thank you!
>
> ```
> Furthermore, the payment system (lines 90–93) resembles a principal-agent task-assignment problem, where a principal contracts an agent to act on their behalf to achieve optimal outcomes. This game has been extensively studied in economics and game theory literature [2], yet it is not acknowledged in this paper.
> ```
>
> Thank you for clarifying! We believe we have acknowledged this connection to the principal-agent contracting game. The acknowledgement is in the lines 90-93 that you noted: "This is often used to model a principal contracting an agent to act on their behalf, with the trajectory informing how much the leader will be paying ex-post" (L92-93).
>
> In light of your comment, we will be sure to make this more clear, and include a pointer to the broader algorithmic contracting literature (e.g. [2]) as you suggest.
>
> ```
> The authors' perspective is interesting, though some similar concepts may already exist in the field.
> ```
>
> Thanks for this! We believe a strength of our paper is the generality of the setting we study, which subsumes some of the more specific settings studied in prior works. For instance, one general result of ours is an efficient algorithm for learning with payment (which we show to be necessary) in *any* Cooperative Stackelberg Markov Game.
>
> ```
> To clarify further, regarding the KKT reformulation of bilevel optimization problems [3]—a well-known and elegant methodology in bilevel and multi-level optimization—there are known limitations. However, some bilevel optimization problems can be formulated as linear programs (LPs) or convex quadratic programs (QPs), leading to a poly(T) solution. Thus, I am uncertain about the author's claims of NP-hardness (though I could be mistaken). Contrary to the authors' rebuttal, this is a global solution (not local). Additionally, gradient ascent algorithms for bilevel optimization can lead to sufficient local solutions, but I am unsure if this was covered in the literature review [4,5]—it seems relevant here.
> ```
>
> Thank you very much for helping us understand your point!
>
> 1. As you write, *some* bi-level optimization programs can be formulated as a LP/QP, whose global optimum can be computed in poly time. We agree with this. But would you agree that this statement does not imply *all* bi-level optimization programs can be solved in poly time?
>
> 2. Our result is that the optimization program in the general-sum case cannot be solved in poly time. We believe this does not contradict your statement that *some* bi-level programs are solvable in poly time.
>
> Please let us know if this helped to address your uncertainty about our NP-hardness result. And thank you for your references [4] and [5]! We will be sure to include our discussion on bi-level optimizers and local vs global optimum in the revision.
>
> ```
> I appreciate the authors' inclusion of additional empirical results. Given the paper's claims, empirical validation is essential to strengthen its credibility and applicability. Overall, I believe the clarity of the work needs improvement. I found the formulations and proofs difficult to follow—either due to my own failure to appreciate the work or because the paper requires further refinement.
>
> I'll updated my score in light of the rebuttal and have no further questions.
> ```
>
> Thank you for this note on the clarity of our paper, and for updating your score!
>
> We will work to improve our paper using your detailed feedback. Please let us know if you have any further questions, thank you!

---

### Official Review · Reviewer_twSH · 2025-07-01

**Clarity:** 3
**Significance:** 3
**Originality:** 3
**Rating:** 5
**Confidence:** 2

**Summary:**

This paper analyses the computational hardness of Stackelberg games, with a especial emphasis on the setup where the leader is able to influence the policy of the follower by modifying its rewards (a.k.a. "payment") at the trajectory level (adding a reward for each state-action pair on the realized trajectory) or upfront (before the trajectory is realized). The authors study three main settings of Stackelberg games and provide the following computational hardness results on each of them:

1. General-sum Stackelberg games: An efficient algorithm that returns both optimal policy and payment cannot exist unless $\text{P}=\text{NP}$.
2. Cooperative Stackelberg games: An efficient algorithm that returns both optimal policy and payment cannot exist due to statistical hardness tied to structural properties of the MDP.
3. Cooperative Stackelberg games with payment: The authors introduce an efficient algorithm, UCB-VI with Follower Payment (UCB-VI-FP), for which they prove upper and lower regret bounds for the trajectory level and upfront payment setups.

**Questions:**

1. Is there a simulator or game that could be used to demonstrate the effectiveness of UCB-VI-FP empirically? Is this a direction that the authors have considered or is the algorithm mostly a theoretical construct for the hardness proofs?

2. How do you anticipate the results to change when considering $n$ agents instead of two? The *curse of multi-agents* is well known in related literature and I wonder if there is a hope that an extension to UCB-VI-FP could also be efficient in the $n$-player setting.

3. Are there any other real-world examples of two player cooperative Stackelberg games with payment (besides the AI assistants example discussed in the paper) that could benefit from the efficiency results?

**Ethical Concerns:**

["NO or VERY MINOR ethics concerns only"]

**Final Justification:**

My recommended score remains the same, I believe in the practical utility of this paper and the discussion helped me improve my confidence.

**Limitations:**

The Discussion section contains a candid “Limitations” paragraph that flags the full-information assumption, namely, that the leader can observe both the follower’s actions and rewards. It also notes that this may fail in physical settings or where rewards are privately reported.

**Paper Formatting Concerns:**

No formatting concerns.

**Quality:**

3

**Strengths And Weaknesses:**

I want to preface by stating that I am not an expert on reinforcement learning theory, therefore I am unable to validate the correctness of the proofs presented on this paper. Everything on this review assumes that the theoretical results are correct and well justified.

## Strengths

**Clarity:** The paper is well written. In particular, the theoretical results are explained and discussed in detail, to the extend that a non-expert like myself can understand them and their implications. The mathematical notation is consistent and unambiguous, as each artifact is defined and sometimes explained.

**Novelty:** The study of the computational hardness of Stackelberg games in the three settings discussed on the paper is completely novel. The authors also introduce a new algorithm based on UCB-VI-FP to prove that cooperative Stackelberg games with payment can be solved efficiently and compute its corresponding regret bounds in two different settings.

**Impact:** Stackelberg games are a model that can be used in all sorts of interaction, therefore the results about the computational hardness of these games are of great interest to both experimentalists and theoreticians in areas like game theory, multi-agent reinforcement learning, economics, among others.

## Weaknesses:

**Lack of experimental results:** The paper’s contributions are entirely theoretical, leaving readers without empirical evidence that the proposed payment schemes can be implemented stably or efficiently in practice. Even a small-scale simulation or toy example would support the validity of the theoretical results and overall strengthen the paper.

**Applicability to real world problems:** In this particular case I am not referring to Stackelberg games in general, but the particular subset of Stackelberg games for which the authors prove that an efficient algorithm exist. Namely, two player cooperative games with payment seem to be very limited in scope and not directly tied to real world applications.

---

> ### Author Rebuttal · Authors · 2025-07-31
>
> Thank you very much for your detailed review of our paper, Reviewer twSH! We appreciate your useful feedback that has helped to improve our paper.
>
> ```
> Lack of experimental results: The paper’s contributions are entirely theoretical, leaving readers without empirical evidence that the proposed payment schemes can be implemented stably or efficiently in practice. Even a small-scale simulation or toy example would support the validity of the theoretical results and overall strengthen the paper.
> ```
>
> Thanks for this note! Indeed, our contributions in this paper are mainly theoretical. Theory is important in this setting as algorithms are used to compute payment. Thus, provable guarantees are important to ensure that the payment computed are sensible/optimal (L39-40).
>
> With this said, we have performed experiments, following your feedback that experiments would further improve the paper. Please find below a description of the experiment we ran.
>
>
> **Setup:** We consider a turn-based Markov Game, where the leader is solving a RL problem and the follower solves a bandit problem in its BR (Bai et al). This class of Markov Games includes the hard instance construction in Section 5. Our goal is to examine whether learning without payment can get stuck in more "average (and not worst) case" Markov Games.
>
>
> The leader is doing RL in a toy MDP with $H = 5$. For the follower, arm $a_1$ leads to a MDP, whose optimal return is the optimal return in the (turn-based) Markov game. Arm $a_2$ has a deterministic high reward that is $\alpha$ that of the optimal return.
>
>
> By varying $\alpha$, we can make the follower get "stuck" in myopically choosing $a_2$, thus preventing the leader from exploring and learning the actual optimal policy. This is the intuition behind the negative results, Theorem 5.1 and 5.2, where we set $\alpha$ very high.
>
>
> For the baseline, we use the single-agent learning algorithm UCB-VI in the without payment case, and compare it against UCB-VI-FP in the with payment case. We track the cumulative regret of the two learning algorithms over $40000$ episodes and across $20$ runs.
>
>
> **Finding:** We experiment with different $\alpha$’s, finding that learning without payment can get stuck in the myopic optimum even when $\alpha$ is as low as $0.5$. Interestingly, this suggests that there are “non-worst-case” Markov games, where exploration can be difficult without payment.
>
>
> Under $\alpha = 0.5$, our plots show that:
>
> 1. In absence of payment, even if the leader is using a (one-sided) no-regret algorithm (UCB-VI), the leader may not be able to explore adequately and incur linear regret.
> 2. UCB-VI-FP attains sublinear regret, showing the importance of payment in incentivizing  exploration needed to learn the optimal policy.
> 3. UCB-VI-FP initially incurs a higher regret, which we expect due to the additional payment used by the algorithm to incentivize exploration. Over time, its regret improves due to exploration shrinking the policy regret, and reduced incentivization and thus payment regret. UCB-VI-FP’s regret eventually dips below that of UCB-VI, with a crossover point at around episode 13500.
> 4. UCB-VI-FP’s regret converges like $O(\sqrt{K})$, which matches our upper bound. We verify this upper bound for every episode $i$ for $i \in [40000]$.
>
> Here is a table summarizing average regret (cumulative regret / $T$) across episodes, averaged over the runs.
>
> | Episode $T$  | $5000$ | $10000$ | $15000$  | $20000$ | $25000$ | $30000$ | $35000$ | $40000$ |
> |--------------|--------|---------|----------|---------|---------|---------|---------|---------|
> | UCB-VI Average Regret  | 0.42   | 0.42    | 0.42     | 0.42    | 0.42    | 0.42    | 0.42    | 0.42    |
> | UCB-VI-FP Average Regret  | 0.70 | 0.49    | 0.40     | 0.35    | 0.31    | 0.28    | 0.26    | 0.24    |
>
>
> ```
> Applicability to real world problems: In this particular case I am not referring to Stackelberg games in general, but the particular subset of Stackelberg games for which the authors prove that an efficient algorithm exist. Namely, two player cooperative games with payment seem to be very limited in scope and not directly tied to real world applications…Are there any other real-world examples of two player cooperative Stackelberg games with payment (besides the AI assistants example discussed in the paper) that could benefit from the efficiency results?
> ```
>
> Firstly, to clarify, Cooperative Stackelberg Markov games with payment are not limited in scope. This class of games generalizes Cooperative Stackelberg Markov games (without payment). Indeed, constraining the leader to zero payment (i.e. $(\pi, b) = (\pi, 0)$) corresponds to the leader’s policy space in Cooperative Stackelberg Markov Games (without payment).
>
> As we write in L77-78, payment is simply an *additional* action that the leader can take to enhance cooperation. And more generally, Stackelberg Markov game with payment generalizes Stackelberg Markov games, which is a broad class of games as you noted earlier.
>
> Secondly, in terms of applicability to real world problems, two-player Cooperative Markov games is a general class of games that models cooperative real world settings, wherein states are fully observable. These include human-AI collaboration, where human is the leader and AI is some digital and/or physical assistant agent, and AI-AI collaboration, which can include interaction among digital and/or physical agents representing different businesses.
>
> We believe that our work is most relevant in settings involving contracting and/or interaction between agents (be they human or AI) representing two businesses. In these *decentralized* settings, payment becomes a natural (economic) means to induce cooperation, as joint training is not possible. As we write in our introduction, we believe we will soon see more of such settings arise, with agents becoming more capable and more widely deployed by businesses.
>
> ```
> Is there a simulator or game that could be used to demonstrate the effectiveness of UCB-VI-FP empirically? Is this a direction that the authors have considered or is the algorithm mostly a theoretical construct for the hardness proofs?
> ```
>
> Yes, please see the experiment description we have provided earlier, which detail our setup for empirically demonstrating the effectiveness of UCB-VI-FP. And yes, you’re right that our algorithm also serves as a theoretical upper bound on regret that complements the hardness results.
>
>
> ```
> How do you anticipate the results to change when considering n agents instead of two? The curse of multi-agents is well known in related literature and I wonder if there is a hope that an extension to UCB-VI-FP could also be efficient in the n-player setting.
> ```
>
> Thanks for this question! One consideration is that with $n$ agents, the action space becomes exponential in $n$. And so, we anticipate that additional assumptions (e.g. factorization assumptions) are needed so that positive results could be attainable.

---

> > ### Comment · Reviewer_twSH · 2025-08-01
> > **Reply to rebuttal**
> >
> > I would like to thank the authors for clarifying my questions and incorporating the feedback that I initially provided. In particular, I am feeling more confident on my initial assesment now that 1) I have been made aware that Cooperative Stackelberg Markov games with payment are more general than Cooperative Stackelberg Markov games and 2) I can empirically see the claims being validated in the toy example. Therefore, I believe that this paper has a substantial area of applicability for a broad section of empiricists like myself who work in related areas.

---

### Official Review · Reviewer_fS36 · 2025-07-02

**Clarity:** 3
**Significance:** 3
**Originality:** 3
**Rating:** 5
**Confidence:** 3

**Summary:**

Setting:

- This paper considers the computational and statistical complexity of leader-follower (Stackelberg) games. In these games, there are two players and a series of stages called epochs. In each epoch, the leader commits to a strategy and then the follower, observing the leader’s policy, commits to its own strategy. Over the course of a epoch, the current state, leader’s action and follower’s action mutually determine the next state. Each leader and follower receives a (possibly differing) reward based on the trajectory of state-action combinations that was realized. The follower is assumed to best-respond to the leader so that it maximizes its reward under the leader’s policy.
- The paper considers planning problems, where the leader tries to determine the policy/payment that achieves the highest reward, and learning problems, where the goal is the same but the leader does not know the transition or reward functions.
- Many variants of the problem are considered, including:
    - Whether or not to allow the leader to transfer payments to the follower.
    - If payment is allowed, whether the leader must pay “upfront” for all state/action combinations it would like to incentivise or whether the leader just has to pay the follower for the state/action combinations that were realized over the course of the dynamics.
    - Whether the game is general sum, where the leader and follower may have different reward functions, or cooperative, where the leader and follower have the same reward function.
    - Whether the state space is described by a directed acyclic graph (DAG) or a tree.
    - Whether the transition function (which maps the current state and the player’s actions to the next state) is deterministic or stochastic.

Main results:

- Prop 4.1 and Thm 4.2: In general sum games, if the state space is a DAG and transitions are deterministic, with and without payment, it is NP-hard to compute the optimal leader policy (and payment, where applicable).
- Prop 4.3: In the same settings as above, if the state space is instead a tree, planning and learning are computationally/statistically efficient.
- Thm 5.1, 5.2 and Prop 5.3: In cooperative games without payment, if the state space is a tree and transitions are deterministic, planning and learning are efficient. If either condition (tree or deterministic) are relaxed, no statistically efficient learning algorithm exists.
- Thm 6.4 and 6.5: Sublinear regret guarantees for trajectory and upfront payment in cooperative games with payment.

**Questions:**

In Section 6.2, unless I missed it, you did not define indicator bonus? What is this?

**Ethical Concerns:**

["NO or VERY MINOR ethics concerns only"]

**Final Justification:**

This is a clearly written, technically strong paper. My limited concerns were addressed during the rebuttal period, and I did not see any concerns raised by other reviewers that would have led me to revise my scores downward.

**Limitations:**

Yes

**Quality:**

4

**Strengths And Weaknesses:**

Strengths:

- The problem of learning/planning in leader/follower contexts is well-motivated, particularly in contexts with LLM agents etc.
- The theoretical results seem complete to me, with satisfactory answers about the computational and statistical complexity for each of the variants of the game.
- I really appreciate the intuition provided for the results. The discussion around the results is very informative. In particular, around Theorem 5.4, it is interesting that the role of payment is to incentivise the follower to explore actions they would not otherwise take, thereby letting the leader determine the value of such less-desirable states.

Weaknesses:

- There should be a definition of planning versus learning problems in the preliminaries.

---

> ### Author Rebuttal · Authors · 2025-07-31
>
> Thank you very much for your thorough review of our paper, Reviewer fS36! We will be sure to include a definition of planning versus learning in the preliminaries section, as you suggest.
>
> ```
> In Section 6.2, unless I missed it, you did not define indicator bonus? What is this?
> ```
>
> Thanks for this note, we will make this clear in the revision! We mean that the optimistic reward is set to be $\hat{r}(s,a,b) = 1(\text{if $(s,a,b)$ is unvisited})$ and $r(s,a,b)$ otherwise (L1106 in the Appendix).

---

> > ### Comment · Reviewer_fS36 · 2025-08-02
> > **Rebuttal response**
> >
> > Thanks! This addresses my comments. Planning to leave my (positive) review as is.

---

> > > ### Author Response · Authors · 2025-08-04
> > > **Reply to Reviewer fS36**
> > >
> > > Thank you for your timely reply and detailed review, Reviewer fS36! We appreciate it.

---

### Official Review · Reviewer_tKfe · 2025-07-03

**Clarity:** 3
**Significance:** 3
**Originality:** 3
**Rating:** 4
**Confidence:** 2

**Summary:**

This paper studies how to design incentives in long-horizon settings where a principal agent (the “leader”) interacts with another agent (the “follower”) with potentially misaligned interests. The interaction is modeled as a Markov game, and the leader can influence the follower’s behavior by committing to a policy and a payment scheme—either based on realized trajectories or upfront across all state-action pairs. The authors analyze both the planning and learning problems in this setting. They show that finding optimal policies can be computationally or statistically hard in general—particularly in deterministic DAG MDPs for computational hardness, and under probabilistic transitions (even in trees) for statistical hardness—but becomes tractable in certain structured cases, like deterministic tree-structured environments. In the cooperative setting, they propose a UCB-based learning algorithm that leverages payment schemes to achieve sublinear Stackelberg regret. A key insight is that while payments are unnecessary for planning in fully aligned environments, they can be critical for efficient exploration during learning. The paper also draws a contrast between trajectory and upfront payments, showing that the ability to pay adaptively—only for visited states—can substantially improve regret guarantees.

**Questions:**

1. Several of the results (e.g., the hardness in DAGs, the benefit of optimism-based exploration in cooperative settings) felt natural or familiar. Could the authors clarify the extent to which these results are new, and how they relate to prior work in Stackelberg planning, incentive design in MDPs? A more explicit positioning against existing approaches—especially for the learning results—would help assess the originality of the work.

2. In the deterministic tree planning algorithm, a subroutine named “Nash VI” is used to compute a worst-case policy for the follower. However, it was unclear whether this refers to a known algorithm, a construct introduced earlier, or a definition given only in the appendix (I couldn't find other references to it in the appendix). Could the authors clarify its meaning and role in the main text?

3. In the cooperative learning setting with payments, the paper argues that Stackelberg regret can be decomposed into policy regret and payment regret. While the high-level intuition is discussed, I found the mechanics of this decomposition somewhat opaque. A more detailed explanation (perhaps with a figure or simple example) could make the logic clearer.

4. The comparison between upfront and trajectory payments was concrete and insightful. However, I was left wondering about the practical scope: in what types of real-world systems is upfront payment a genuine constraint? Are there more examples from digital platforms, robotics, or AI agent systems where the upfront model is necessary or preferred?

**Ethical Concerns:**

["NO or VERY MINOR ethics concerns only"]

**Final Justification:**

After considering the rebuttal, the author responses, and the discussion with other reviewers, I have improved my evaluation of the paper. The authors’ clarifications addressed many of my earlier concerns, particularly regarding the technical contributions of the learning results and the novelty of hardness results and the UCB-based approach for achieving sublinear Stackelberg regret. The rebuttal also helped clarify the role of the “Nash VI” subroutine and the decomposition of Stackelberg regret, which were points of confusion in my initial reading.

Some minor issues remain—such as a deeper exploration of the practical applicability of the payment models and how the results extend to large-scale environments (especially in number of agents as others points out in reviews). These remaining issues are minor compared to the paper’s strength (clear writing, justified setting, strong theoretical framework) so combined with the clarifications during the discussion, I have raised my score.

**Limitations:**

The authors have made a good effort to acknowledge the limitations of their work, particularly in highlighting assumptions about full observability of follower actions and rewards, as well as the open question of handling misreporting. These are important caveats, especially for applying the framework in real-world or strategic settings. One area that could benefit from further discussion is the practical scope of the payment models—in particular, how common or feasible trajectory vs. upfront payment schemes are in deployed multi-agent systems.

Another limitation not directly addressed is the scalability of the approach in environments with large or continuous state/action spaces. The framework assumes access to full policy and payment representations, but real-world systems may require function approximation or compact representations to be tractable. Discussing whether the planning and learning results extend to such approximate or parameterized settings would strengthen the paper’s practical relevance.

**Paper Formatting Concerns:**

1. At the end of Section 4 it would be helpful to reference where in the appendix local Stackelberg optima are treated
2. I believe the index of the sum in Definition 6.1 should be k

**Quality:**

3

**Strengths And Weaknesses:**

The paper tackles an increasingly relevant question in multi-agent systems: how a principal can design payment schemes to influence another agent’s behavior over long time horizons, particularly in cooperative and partially aligned settings. One strength of the paper is its clear structure and organization—the distinction between planning and learning settings, and between trajectory and upfront payments, provides a useful framework for understanding the design space. The formalism is clean, and the core contributions are communicated clearly, especially in the cooperative learning setting with payments. The use of optimism-based (UCB-style) algorithms to obtain sublinear Stackelberg regret is an interesting application of existing tools in a Stackelberg learning context.

The main theoretical contributions are well-articulated, though it’s somewhat difficult for me to gauge how novel or surprising they are within the broader literature on learning in Markov games or incentive design. Some results, like the hardness in DAG-structured environments and the benefit of trajectory-based payments during exploration, seemed fairly expected given the setup. The use of optimism-based learning (e.g., UCB) is clean and sensible, but may largely follow existing paradigms. Still, the conceptual clarity around payment structure and its impact on regret is helpful and potentially useful for future work in applied settings.

---

> ### Author Rebuttal · Authors · 2025-07-31
>
> Thank you very much for your careful reading of our paper, Reviewer tKfe! We will be sure to include further discussion on the practical scope of the payment schemes and large state/action space settings in Section 7, as you suggest.
>
> ```
> Several of the results (e.g., the hardness in DAGs, the benefit of optimism-based exploration in cooperative settings) felt natural or familiar. Could the authors clarify the extent to which these results are new, and how they relate to prior work in Stackelberg planning, incentive design in MDPs? A more explicit positioning against existing approaches—especially for the learning results—would help assess the originality of the work.
> ```
>
> To clarify, we believe all of our results are novel. In terms of prior work, there are a few papers that study Stackelberg Markov games with payment (L133-145). All of these papers consider the setting where the leader can only set payments (but not policy), which is a special case of the setting we consider. To our knowledge, our paper is the first to study Stackelberg Markov games with payment, wherein the leader can set both the policy and the payment.
>
> 1. For the hardness in DAGs result that you reference, we note that it does not hold in the less general setting; the leader would simply set the optimal payment to be zero. And so, our negative result is novel as it only arises in the more general setup, which we study.
> Moreover, we note that planning is efficient in cooperative games. And so, at least we found it surprising that learning can be statistically intractable, even when the rewards are *already* aligned (L215-216).
>
> 2. For the positive result (UCB-VI-FP) that you reference, we believe this is novel as well and complements the (new) negative result discussed above. Indeed, while multi-agent RL algorithms where agents jointly explore are common, multi-agent algorithms where only one agent can explore are less common and require addressing new challenges. To see this, Theorem 5.1, 5.2 show that even if the leader is using a no-regret learning algorithm, then learning can still be inefficient.
> And so, a new learning algorithm needs to be developed here, using payment to incentivize collective exploration. The challenge in algorithm design is two-fold. First, the payments have to induce joint exploration to lead to sublinear policy regret. Second, and importantly, the payments also need to be strategically set so that the total payment does not grow arbitrarily, but rather tapers off to be sublinear in $T$. Controlling total payment growth along with policy regret is the new challenge we have to handle in this setting.
>
> 3. Furthermore, we add that we also obtain results under upfront payment, which is a new form of payment that has not been considered in prior literature.
>
> Overall, we are confident that all of our results are novel. Certainly, Stackelberg games is a very well studied topic. So please let us know if there are any other results we could comment on to help assess the originality of our work, thanks!
>
> Lastly, we wish to point out that a main strength of our paper, in addition to the clear organization that you mention, is the comprehensiveness of our set of positive and negative results. They serve to tightly show where the complexity barriers are in this general setup.
>
>
> ```
> In the deterministic tree planning algorithm, a subroutine named “Nash VI” is used to compute a worst-case policy for the follower. However, it was unclear whether this refers to a known algorithm, a construct introduced earlier, or a definition given only in the appendix (I couldn't find other references to it in the appendix). Could the authors clarify its meaning and role in the main text?
> ```
>
> In RL theory, Nash Value Iteration (Nash VI) is a known planning algorithm (mentioned e.g. in Bai et al) that computes the minimax optimal policy in zero-sum Markov games. We will be sure to include a reference to it in the revision. Thanks for this note!
>
> ```
> In the cooperative learning setting with payments, the paper argues that Stackelberg regret can be decomposed into policy regret and payment regret. While the high-level intuition is discussed, I found the mechanics of this decomposition somewhat opaque. A more detailed explanation (perhaps with a figure or simple example) could make the logic clearer.
> ```
>
> This follows from that the value function is additive in terms of rewards:
> $V^{\pi, \mu}(s_0; r - \kappa \cdot b) = V^{\pi, \mu}(s_0; r) - V^{\pi, \mu}(s_0;  \kappa \cdot b)$, where the first term is the return and the second is the payment incurred. Please let us know if this has addressed your question, thanks! Overall, we agree with you that proof sketches would benefit the presentation. Please understand our paper was fairly constrained in terms of space in order to fit all the results in.
>
>
> ```
> The comparison between upfront and trajectory payments was concrete and insightful. However, I was left wondering about the practical scope: in what types of real-world systems is upfront payment a genuine constraint? Are there more examples from digital platforms, robotics, or AI agent systems where the upfront model is necessary or preferred?
> ```
>
> We believe that there are genuine constraints in the real world such that upfront payment is preferred and/or necessary.
>
> Another example is insurance, which is another form of upfront payment. For example, a company that has leased a robot may wish to purchase insurance upfront for the robot. Paying for the insurance means paying some fee upfront to be insured against bad scenarios (states) that the robot could land in during use. And verily these scenarios may not all be realized ex-post.
>
> More generally, thanks to your question, we ended up finding a sizable body of economics contracting literature where the principal can only pay the agent ex-ante (i.e. upfront payment). Some reasons for this are:
>
> 1. Non-enforceable/costly contract situations: when the principal can renege upon observing the outcome.
> 2. Non-verifiable outcomes situations: when outcomes cannot be verified by third parties, ex-post contracts become unenforceable (Aghion and Holden). So even if both parties can observe the outcome, without verifiability, there's no way to condition legally binding payments on results. This makes ex-ante payments necessary.
> 3. Risk-aversion on the part of the agent: when the agent is risk-averse, upfront payment may be preferred when the outcome is stochastic.
>
> While we are not sure what future business settings are like with digital agents, based on this line of economics literature, it does seem like there are a number of reasonable assumptions under which upfront payment is preferable. Thus, it seems important to also study upfront payment as a viable form of payment. Thanks again for your question, and we will be sure to include these points when motivating the consideration of the upfront payment!
>
> ```
> At the end of Section 4 it would be helpful to reference where in the appendix local Stackelberg optima are treated
> ```
>
> Thanks for this! Yes, we will be sure to include a pointer to section A.2.2 of the Appendix in the revision.

---

> > ### Comment · Reviewer_tKfe · 2025-08-05
> >
> > Thank you for your thoughtful responses. This has helped me contextualize the novelty of the work more and after reading the response and other reviews I feel more positive about the work.

---

> ### Author Response · Authors · 2025-08-05
> **Reply to Reviewer tKfe**
>
> Thank you for getting back to us, Reviewer tKfe! Stackelberg games are a general class of games with an extensive body of literature. So it makes a lot of sense that you had some questions on how our work is positioned w.r.t. existing works.
>
> We are glad to hear that we are able to address the questions you had. Please do let us know if you have any further questions. Thank you once again for your review, which has helped to improve our paper and better contextualize the novelty of our results!

---

### Author Response · Authors · 2025-08-09
**Thank you for your reviews and discussions**

To conclude, we thank all the reviewers for their constructive reviews, which have greatly helped to improve our paper! We are glad that the reviewers found our formulation well-motivated (Reviewers tKfe, fS36, twSH, jtyD), our presentation clear (Reviewers tKfe, fS36, twSH) and our results interesting (Reviewers tKfe, fS36), useful (Reviewers tKfe, twSH) and complete (Reviewers fS36).

For accountability, we document the main changes to our paper in light of the helpful feedback provided by the reviewers. We will be sure to incorporate *all* the feedback given, but for brevity, we list the three key additions here:

1. We will add the experiments in support of our theoretical results.
2. We will expand on the generality of the game we study.
3. We will further clarify how our paper is positioned within the broader Stackelberg game literature, including comparisons with existing works on contracting in MDPs (from discussions with Reviewer tKfe) and comparisons with bi-level optimization algorithms and of local vs global optimum (from discussions with Reviewer jtyD).

We will work to add all these points in our next revision. In closing, we thank all the reviewers again for taking the time to review our paper!

---

### Decision · Program_Chairs · 2025-09-17

**Decision:**

Accept (poster)

**Comment:**

This paper studies Stackelberg Markov games with outcome-based payments, offering a theoretical treatment of planning and learning under different reward alignment conditions. The authors establish hardness results for general-sum settings and identify structural cases where efficient solutions exist. They also introduce a UCB-based learning algorithm for cooperative settings with payments, proving sublinear Stackelberg regret and clarifying the necessity of payments even with aligned rewards. Strengths include the clean formalization, thorough characterization of complexity boundaries, and clear exposition. While the main weakness lies in the absence of extensive empirical validation and some questions regarding scope and scalability, the authors partially addressed this with toy experiments and clarifications during rebuttal and the authors' final comment. Overall, the combination of rigorous theoretical contributions, strong motivation, and responsive engagement with reviewer concerns justifies acceptance.